# DepthCloak: Projecting Optical Camouflage Patches for Erroneous Monocular Depth Estimation of Vehicles

## ABSTRACT

Adhesive adversarial patches have been common used in attacks against the computer vision task of monocular depth estimation (MDE). Compared to physical patches permanently attached to target objects, optical projection patches show great flexibility and have gained wide research attention. However, applying digital patches for direct projection may lead to partial blurring or omission of details in the captured patches, attributed to high information density, surface depth discrepancies, and non-uniform pixel distribution. To address these challenges, in this work we introduce DepthCloak, an adversarial optical patch designed to interfere with the MDE of vehicles. To this end, we first simplify the patch to a gray pattern because the projected "black-and-white light" has strong robustness to ambient light. We propose a generative adversarial network (GAN) based approach to simulate projections and deduce a projectable list. Then, we employ neighborhood averaging to fill sparse depth values, compress all depth values into a reduced dynamic range via nonlinear mapping, and use these values to adjust the Gaussian blur radius as weight parameters, thereby simulating depth variation effects. Finally, by integrating Moiré pattern and applying style transfer techniques, we customize adversarial patches featuring regularly arranged characteristics. We deploy DepthCloak in real driving scenarios, and extensive experiments demonstrate that DepthCloak can achieve depth errors of over nine meters in both bright and night-time conditions while achieving an attack success rate of over 80% in the physical world.

## CCS CONCEPTS

• **Security and privacy** → *Systems security*.

## KEYWORDS

Monocular Depth Estimation, Optical Projection Attack, Physical Attack

## 1 INTRODUCTION

In comparison with stereovision (employing ≥ two cameras) [30, 31] and laser-based depth perception technologies (e.g., LiDAR) [13], monocular depth estimation (MDE) [15] has quickly emerged as an essential technology in autonomous driving or assisted driving systems due to its cost-effectiveness, simplicity, and ease of integration. Numerous automotive manufacturers have incorporated

Permission to make digital or hard copies of all or part of this work for personal or classroom use is granted without fee provided that copies are not made or distributed for profit or commercial advantage and that copies bear this notice and the full citation on the first page. Copyrights for components of this work owned by others than the author(s) must be honored. Abstracting with credit is permitted. To copy otherwise, or republish, to post on servers or to redistribute to lists, requires prior specific permission and/or a fee. Request permissions from permissions@acm.org.

*ACM MM, 2024, Melbourne, Australia*

© 2024 Copyright held by the owner/author(s). Publication rights licensed to ACM.
ACM ISBN 978-x-xxxx-xxxx-x/YY/MM
https://doi.org/10.1145/nnnnnnn.nnnnnnn

MDE into their perception systems. For instance, Tesla has recently introduced the utilization of self-supervised models in MDE [24].

MDE predominantly acquires relative depth information of a scene indirectly through the analysis of color, vertical object arrangement, shadows, and other visual cues. However, a solitary two-dimensional (2D) image does not provide explicit information regarding the three-dimensional (3D) spatial arrangement, and the model dependence on the indirect visual hints is not inherently associated with depth. This interdependence renders MDE particularly vulnerable to adversarial attacks, wherein attackers can easily manipulate image colors, create artificial shadows, or change the positions of objects to disrupt and deceive the depth estimation process.

Attack strategies targeting at depth estimation models can be generally classified into two types: digital perturbation and physical patch-pasting attacks. Zhang et al. [33] demonstrate that digital perturbation attacks introduce perturbations in the digital domain by specifying various attack scenarios for depth estimation including non-targeted, targeted, and universal attacks. Cheng et al. [1] present a physical patch pasting attack, where pre-trained adversarial patches are affixed to targeted areas such as the rear of vehicles to interfere with depth estimation. Despite their effectiveness, limitations still exist. Digital perturbation attacks rely on the assumption that attackers possess comprehensive scene information before the image is processed by the depth estimation system, a condition often unattainable in real-world scenarios. On the other hand, physical patch-pasting attacks suffer from inflexibility, necessitating physical modification of targets, and are susceptible to exposing the attacker's identity through patch recognition.

A recently proposed physical attack technique that has attracted considerable attention is the light projection attack. SLAP [18] proposes a projectable color spectrum and employs a projection model to create adversarial perturbations, which can effectively deceive target recognition systems. OPAD [3] strategically updates adversarial samples by estimating color mixing matrices in the presence of constant illumination gradients. However, such light projection attacks exhibit specific limitations: (1) They mainly focus on compromising target recognition systems (e.g., facial recognition systems, traffic sign recognition); (2) They are particularly susceptible to significant environmental light disturbances; (3) They necessitate precise alignment of projected disturbances with the targeted areas.

To overcome these limitations of light projection attacks, this study introduces a novel MDE attack named DepthCloak, which exhibits high robustness to variations in environmental lighting conditions. Figure 1 depicts an example of the attack scenario. Attackers can remotely manipulate light patches that are projected onto the front of targeted vehicles, leading to the generation of anticipated errors in the victim vehicle's depth estimation system. Note that the direct application of pre-trained adhesive patches

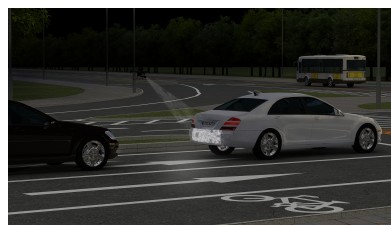

**Figure 1: An example of our attack scenario.**

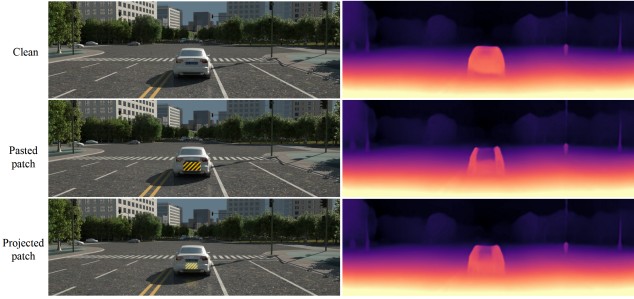

**Figure 2: The attack performance evaluation of the state-of-the-art approach [1] in simulated scenarios using both pasted and projected patches.**

to light projection can cause a decrease in the effectiveness of adversarial attacks as illustrated in the simulated results (Figure 2). Therefore, the creation of an MDE light projection patch withstanding variations in the physical environment still poses a significant challenge. This study undertakes a comprehensive analysis of the causes of failure. The challenges are outlined as follows:

**Challenge 1 (High Visual Information Density):** The high density of visual information within color patches may cause the loss of essential patch details under conditions of intense environmental illumination.

**Challenge 2 (Effect of Non-Planar Surface):** The inherent non-planar surface of the designated target area, accompanied by variations in depth, introduces a significant propensity for the visual distortion of patches.

**Challenge 3 (Disparity in Pixel Distribution):** A heterogeneous distribution of pixels within patches can engender the erroneous transcription of critical information. This challenge is exacerbated by phenomena such as the dispersion of light, suboptimal reflective properties, and the smoothing effect of camera processing algorithms.

We conduct preliminary analysis on the robustness of various colors in intense lighting environments and find that white and black exhibit the lowest sensitivity to environmental light disruptions. Notably, when projected, white and black create optical illusions through contrast manipulation. White is perceived by elevating the brightness of the background, while black is conventionally represented by using the background colors. Therefore, the adversarial patches are first simplified into grayscale. Then, we analyze the absolute alterations in background colors captured by the camera under white light projection across various environmental light

conditions. We identify distinct distribution patterns of brightness and saturation in mixed colors resulting from projecting white light onto various background colors. Based on this discovery, we design a GAN-based simulation of camera-projection processes. The primary focus is to minimize the disparities in brightness and saturation levels between the synthesized projection image and the authentic projection image. Consequently, a projectable list is established to compensate for the projection loss during light patch training.

To mitigate the blurring problem resulting from non-planar projection surfaces, we introduce a strategy for modeling projection depth. While conventional 3D modeling methods can yield precise models, the process is time-consuming and lacks differentiability. To this end, we adopt a technique to streamline the process of 3D modeling into a differentiable 2D image processing procedure. Initially, a simple and effective neighborhood averaging technique is employed to address the problem of missing depth values in the depth map. Subsequently, these depth values are compressed into a reduced dynamic range via nonlinear mapping. The adjusted depth values are utilized as weight parameters to accurately modify the radius of Gaussian blur. This process effectively replicates depth of field effects.

To address the issue of non-uniform pixel distribution in adversarial patches, we utilize deep photo style transfer [19] as the solution. Specifically, opting for two sets of interleaved regular sinusoidal waves creates the foundational design. While maintaining the overall equilibrium of patch content and style, we further enhance and modify the patches by integrating projection losses with the anticipated depth losses in the specified area. This customization effectively tailored adversarial patches with systematically arranged features resembling Moiré patterns.

We evaluate the attack performance of DepthCloak in digital and physical domains on three SOTA self-supervised MDE models (Monodepth2[5], Depthhints[25], and Manydepth[26]). In digital-world testing phase, which encompasses 100 evaluation scenarios, DepthCloak effectively induces the depth estimation root mean square error (RMSE) of more than 15 in 76 scenarios spanning all three target models. During the assessment of physical-world, we observe that in up to 70% of video frames, the RMSE of target area depth estimation exceeds 15. Additional robustness testing reveals that DepthCloak displays enhanced attack performance compared to the other two attack methods, particularly under low-light conditions, showcasing more pronounced performance advantages.

## 2 RELATED WORK

### 2.1 Monocular Depth Estimation Attacks

Recently, numerous studies have investigated the impact of adversarial perturbations on MDE tasks. Zhang et al. [33] refine existing attack methodologies designed for image classification tasks, adapting them for depth estimation tasks. They successfully induced depth estimation discrepancies of 3 to 4 times the real values for specified targets. Wong et al. [28] utilize an iterative optimization approach to find subtle additional perturbations capable of altering predictions made by depth prediction networks (e.g. removing the target while keeping other scene components intact). Mopuri et al. [20] examine universal perturbations in a data-independent setting

for segmentation and depth prediction to alter predictions in arbitrary directions. Dijk et al. [27] believe that most MDE networks overlook the apparent size of known obstacles, the minor camera pitch changes also disrupt estimated distances to obstacles. Hu et al. [9] employ the iterative fast gradient sign method (IFGSM) to disrupt depth estimation CNNs, and propose a defense mechanism based on predicted saliency maps.

However, such a perturbation strategy relies on the assumption that attackers can manipulate the entire scene before the image input system, which is impractical in real-world scenarios. Consequently, adversarial patch attacks have gained significant attention due to their higher feasibility for deployment in physical-world.

Zheng et al. [10] propose a 3D texture adversarial attack against MDE models. They conduct robustness simulation tests under adverse weather conditions such as rainy and foggy. Guesmi et al. [6] introduce a shape-sensitive adversarial patch (SSAP) to disrupt both CNN-based and Transformer-based MDE models. Yun et al. [29] strategically place and generate adversarial patches to deceive MDE models, thereby distorting the depth estimation results of target vehicles. SAAM utilizes data augmentation techniques and introduces semantic constraints to ensure that the generated adversarial patterns visually resemble natural images through a projection function [7].

However, those works are limited to simulating attacks in digital-world and not deploying in physical-world. Yamanaka et al. [32] deploy the patches pre-trained in digital-world into physical-world through printing. They analyze the behavior of MDE under attacks by visualizing the activation levels of intermediate layers and regions potentially affected by adversarial attacks. Cheng et al. [1] balance the stealthiness and effectiveness of attacks through object-oriented adversarial design, sensitive area localization, and natural style camouflage. These works demonstrate effectiveness in physical-world, while they may suffer from the following limitations: 1) Attackers require physical contact with the target object; 2) Printed patches may leave permanent attack traces; 3) Attack performance may decrease in low-light conditions.

## 2.2 Optical Projection Attacks

Optical projection attacks are favored by attackers for their flexible approach to deploying attacks. AdvCP employs particle swarm optimization to search for the physical parameters of color projection, then projecting meticulously designed optical patterns onto specified target objects[8]. Huang et al. [11] utilize PCNet to simulate real-world projection and capture processes with high fidelity. Nguyen et al. [22] explore the feasibility of conducting real-time physical attacks on facial recognition systems through the use of adversarial light projection. OPAD employs structured illumination to alter the appearance of target objects, constructing a form of adversarial attack in physical-world that can effectively deceive image classifiers [3]. Zhong et al. [4] craft physical perturbations based on the natural phenomenon of light and shadow, which is more natural and covert. Li et al. [17] propose a structured light attack targeting 3D facial recognition systems. Zhou et al. [16] exploit vulnerabilities associated with lens flare effects in optical imaging to inject false obstacle depths. SLAP establishes a triadic additive relationship model among surfaces, projections, and camera-perceived images, utilizing light projectors to execute short-duration physical adversarial attacks[18]. Muller et al. [21] develop a systematic process to identify effective attack regions in projector attacks and propose a hijacking attack against Siamese trackers.

## 3 ATTACK MODEL

**Attack Goal.** DepthCloak is designed to deceive the depth estimation system of the victim vehicle, leading to misperceptions of the actual distance to the target vehicle (the vehicle with the adversarial patch), especially misidentifying nearby targets as farther away. Additionally, DepthCloak possesses the following characteristics: 1) No physical modification of the target object; 2) Allowing attackers to adjust attack modes and intensity in real-time, providing strong attack flexibility; 3) Maintaining strong attack performance in nighttime environments.

**Attack Scenarios.** DepthCloak enables attackers to remotely control a UAV equipped with a portable projector, which projects pre-trained patches onto specific areas of the target vehicle. Alternatively, it can also be achieved by pre-deploying roadside devices and remotely adjusting the projection direction based on the actual position of the target vehicle. DepthCloak can be deployed in scenarios where the victim vehicle is waiting at traffic lights or leaving toll booths, traveling in the same lane as the target vehicle, thereby increasing the risk of collision between the victim vehicle and the target vehicle.

**Attacker Capabilities.** Assuming the attacker possesses the following capabilities: 1) Full prior knowledge of the depth estimation model, including its architecture, weights, and training data, enabling further analysis of the potential vulnerabilities through reverse engineering techniques; 2) A certain level of physical access, including the ability to deploy the attack at specific locations; 3) The attacker can swiftly deploy and retract attacks within a brief time window, indicating their extensive experience in operating UAVs.

## 4 DESIGN OF DEPTHCLOAK

### 4.1 Overview

We develop an adversarial optical patch designed to withstand ambient light and induce errors in the MDE of vehicles. Figure 3 illustrates the overall pipeline of DepthCloak. Initially, we identify the target area within the selected attack scenario and segment it into a 4×4 grid to accurately ascertain the blocks where the patch will be positioned [1]. Subsequently, the projection modeling of the patch area modulates pixel blurriness by utilizing depth values as weights. The resulting patch is then seamlessly integrated into the scene, with adjustments made to brightness and saturation based on a predefined projectable list. Following this, we construct the depth loss function, aligning it with the desired depth of the target area to optimize the patch design for effective depth perception deception. Finally, the style of the patch is transferred, guided by the reference image.

### 4.2 Projection Modeling

According to the results from Monodepth2 presented in Figure 2, it is evident that the use of the projected patch has mitigated its adversarial effectiveness to a certain extent. This can be attributed

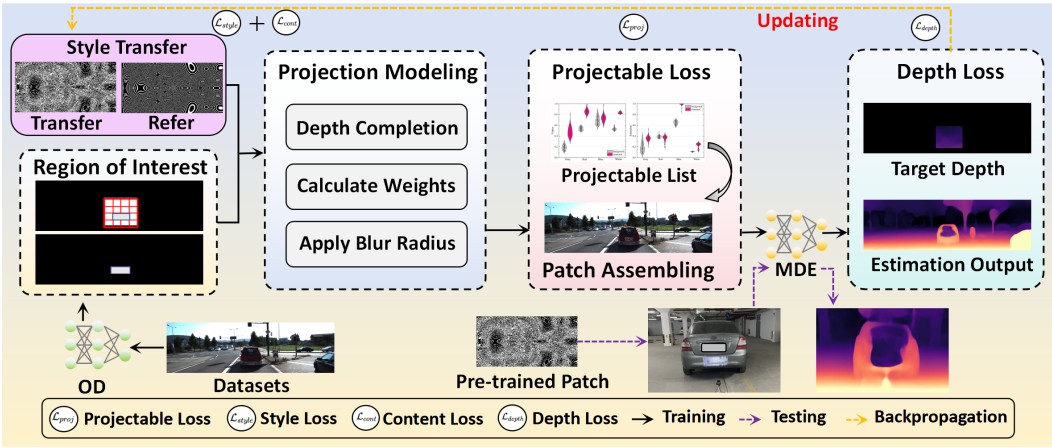

**Figure 3: Overview of DepthCloak.**

to the target surfaces for projection being non-planar, leading to a partial blurring of patch information. DepthCloak develops a depth projection model tailored for projection attacks.

Given a scenario, the target object $O$ and the original patch $p$ are selected, yielding the target mask $m_O$ and the patch mask $m_p$. The sparsity of point clouds obtained by LiDAR leads to the converted depth map having some invalid depth values. Let $d_{i,j}$ be the depth value at pixel location $(i, j)$ in the patch depth map. If $d_{i,j}$ is missing or invalid, we employ the average neighboring pixels to fill it. The completed $d'_{i,j}$ can be formalized as:

$$d'_{i,j} = \begin{cases} d_{i,j}, & \text{if } d_{i,j} \text{ is valid} \\ \frac{1}{|M_{i,j}|} \sum_{(x,y) \in M_{i,j}} d_{x,y}, & \text{if } d_{i,j} \text{ is missing or invalid} \end{cases} \quad (1)$$

where $|M_{i,j}|$ is the number of valid neighboring pixels, and $d_{x,y}$ are the depth values of those valid neighbors.

Then, we incorporate a smooth, nonlinear mapping approach, specifically a Sigmoid function, to compress depth values within a restricted dynamic range of $[0, 1]$. The computation of the weight $\omega_{i,j}$ is delineated as follows:

$$\omega_{i,j} = \frac{1}{1 + e^{-(ad'_{i,j} + b)}}, \quad (2)$$

where $a$ and $b$ are empirically set to -0.5 and 5, respectively.

Finally, we apply the weight $\omega_{i,j}$ to the patch $p$, and perform per-pixel processing using Gaussian blur:

$$p'_{i,j} = \mathbb{G}_{\omega_{i,j}} * p_{i,j}, \quad (3)$$

where the size of $\mathbb{G}$, which influences the level of blur, is directly affected by the weight $\omega_{i,j}$. $p'$ represents the patch obtained after depth modeling.

## 4.3 Projectable Loss

Typically, to enable the effectiveness of adversarial patches in physical-world, attackers employ a non-printable score [23]. The colors captured by the camera during projection are influenced by various factors (e.g., ambient light, background color, and material of the surface). The projection spectrum is much narrower than

that of printed patches under these conditions. We propose a GAN framework to simulate the camera-projection process in the digital domain. The goal is to learn background-projection image mapping to create a projectable list. Figure 4 depicts the GAN framework, comprising the generator and discriminator sub-networks. Our task can be formalized as:

$$\min_G \max_D \mathcal{W}(D, G) = \mathbb{E}_{o \sim p_{proj}} [\log D(o)] \\ + \mathbb{E}_{b \sim p_{bg}} [\log(1 - D(G(b, z)))], \quad (4)$$

where $o$ represents the projected image of the Ground truth, and $b$ stands for the background image. $z$ is a random noise vector sampled from the probability distribution $p_{bg}$.

**Generator Network.** We apply several convolutional layers to extract shallow features of the background image and fuse them with the intermediate features calculated based on the noise vector $z$. This process can be described as follows:

$$o' = G(b, z) = G(b, \mathbf{E}(o)), \quad (5)$$

where $o'$ represents the generated projected image, and $\mathbf{E}$ denotes the encoder. It transforms the projected image into features, i.e., the mean and variance of $z$. The generation loss of $\mathcal{L}_G$ is expressed as:

$$\mathcal{L}_G = \mathcal{L}_{gen} + \alpha_1 \mathcal{L}_z + \alpha_2(\mathcal{L}_{val} + \mathcal{L}_{sat}). \quad (6)$$

$\mathcal{L}_{gen}$ aims to train the generator to output projected images that can deceive the discriminator. can be calculated as:

$$\mathcal{L}_{gen} = log(1 - D(o')). \quad (7)$$

$\mathcal{L}_z$ is used to constrain the latent variable $z$ of the encoder $\mathbf{E}$ to follow an isotropic Gaussian distribution, and is calculated as follows:

$$\mathcal{L}_z = \sum_{i=1}^{n} \left\{ \frac{\mu_i^2 + \sigma_i - log\sigma_i - 1}{2} \right\}, \quad (8)$$

where $\mu$ and $\sigma$ are the mean and variance of $z$, and $n$ denotes the dimensionality of $z$. Brightness loss $\mathcal{L}_{val}$ and saturation loss $\mathcal{L}_{sat}$

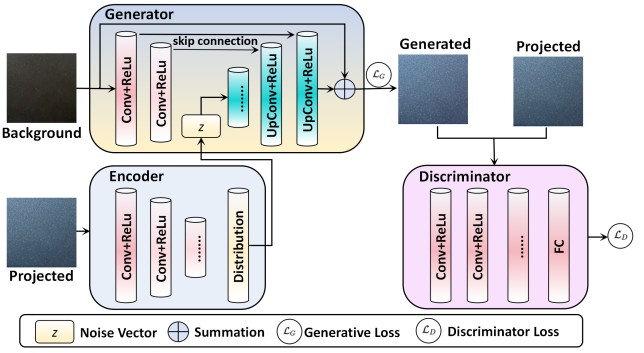

**Figure 4: Camera-projection simulation process based on GAN network.**

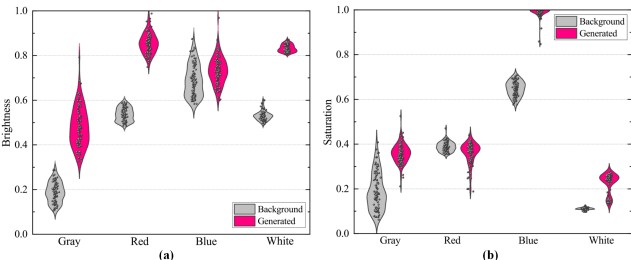

**Figure 5: Brightness and saturation distributions of generated images under different backgrounds.**

can be calculated as $L_2$ norm of $\mathcal{V}$ and $\mathcal{S}$ channels between the generated and real projected images respectively:

$$\mathcal{L}_{val} = \frac{1}{N} \sum_{i=1}^{N} \left\| \mathcal{V}_{gen}^i - \mathcal{V}_{proj}^i \right\|_2, \qquad (9)$$

$$\mathcal{L}_{sat} = \frac{1}{N} \sum_{i=1}^{N} \left\| \mathcal{S}_{gen}^i - \mathcal{S}_{proj}^i \right\|_2, \qquad (10)$$

where $\mathcal{V}_{gen}^i$ and $\mathcal{S}_{gen}^i$ represent the $i^{th}$ pixel brightness and saturation in the generated image, and $\mathcal{V}_{proj}^i$ and $\mathcal{S}_{proj}^i$ in the real image. $N$ is the total pixel count.

**Discriminator Network.** The discriminator $D$ attempts to identify whether the images come from projected images or generated images, with its loss defined as:

$$\mathcal{L}_D = -log(D(o)) - log(1 - D(o')). \qquad (11)$$

To strike a balance between performance and efficiency in our GAN network, we employ a single global discriminator, primarily based on the structure of AlexNet [14]. We can effectively harness its capacity for feature extraction and discrimination while maintaining computational efficiency.

Figure 5 clearly illustrates that for different background colors, there are distinguishable intervals of brightness and saturation. Subsequently, according to the feature extraction rules of the GAN network, we assess the ranges of projected brightness and saturation values for different background colors, used to generate a projectable list $C = (C_v, C_s)$. The projection loss is expressed as:

$$\mathcal{L}_{proj} = \sum_{i=1}^{m} \sum_{j=1}^{n} min \left\| \mathcal{V}_{p_{i,j}} - C_v \right\|_2 + \left\| \mathcal{S}_{p_{i,j}} - C_s \right\|_2, \qquad (12)$$

where $\mathcal{V}_{p_{i,j}}$ and $\mathcal{S}_{p_{i,j}}$ represent the brightness and saturation of patch pixels, respectively.

### 4.4 Style Transfer

To mitigate the interference from strong ambient light on the projected patch, we simplify the patch's color complexity and restrict it to a gray pattern. This decision stems from projection technology principles: perceived white light enhances background brightness, while "black light" is a visual illusion caused by contrast, not actual

light. Empirical testing has identified two main projection strategies: white projection (adjusting background brightness) and black projection (preserving the original background color).

Unfortunately, while gray patches mitigate strong ambient light, their non-uniform pixel distribution causes camera-captured images to lack essential information. Consequently, we introduce Moiré patterns (formed by the crossing of two waveforms, distributed uniformly) as the reference style, and employ deep photo style transfer [19] for patch customization. Specifically, we define two losses: style loss $\mathcal{L}_{style}$ and content loss $\mathcal{L}_{cont}$, represented as follows:

$$\mathcal{L}_{style} = \sum_{l=1}^{L} \left\| \mathbf{G}(\mathbf{F}_l(p_{style}) - \mathbf{G}(\mathbf{F}_l(p^0)) \right\|_2, \qquad (13)$$

$$\mathcal{L}_{cont} = \sum_{l=1}^{L} \left\| \mathbf{F}_l(p) - \mathbf{F}_l(p^0) \right\|_2, \qquad (14)$$

where $\mathbf{F}_l$ represents the features extracted at the $l^{th}$ layer of the feature extraction network $\mathbf{F}$, and $\mathbf{G}$ is the Gram matrix of deep features. $L$ denotes the count of convolutional layers in $\mathbf{F}$. $p_{style}$ is the style reference image, and $p^0$ is the transferred patch image.

### 4.5 Adversarial Patch Updating

The depth loss is designed as the most critical loss to amplify the estimation error, represented as follows:

$$\mathcal{L}_{depth} = \sum_{i=1}^{m} \sum_{j=1}^{n} \left\| \mathcal{D}_{target} - \mathcal{M}_{p'_{i,j}} \right\|_1, \qquad (15)$$

$\mathcal{D}_{target}$ represents the desired depth of the target area, while $\mathcal{M}$ refers to the depth estimation obtained by applying $p'$ to the target area. Our total loss is denoted as:

$$\mathcal{L}_{total} = \mathcal{L}_{depth} + \alpha \mathcal{L}_{proj} + \beta(\mathcal{L}_{style} + \mathcal{L}_{cont}). \qquad (16)$$

The image processing pipeline is architected to be entirely differentiable, enabling efficient gradient backpropagation from depth estimation to the input adversarial patch. Additionally, we use Adam to optimize the total loss. The objective is to minimize the total loss $\mathcal{L}_{total}$, freezing depth network weights and biases, and updating the adversarial patch via backpropagation.

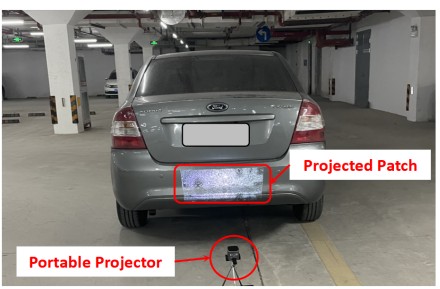

Figure 6: Deployment of DepthCloak in physical-world.

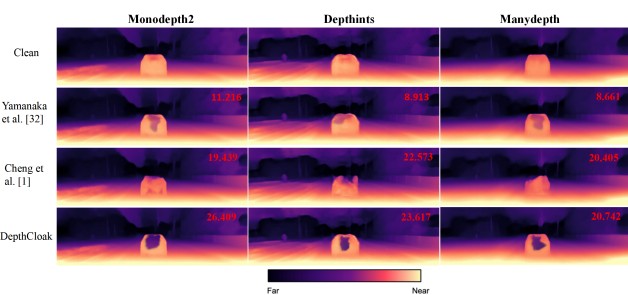

Figure 7: Depth discrepancy under different attacks in digital-world.

## 5 EXPERIMENTS

### 5.1 Experimental Setup

**Model Selection.** In the experiments, three state-of-the-art (SOTA) MDE models are employed: Monodepth2[1], Depthhints[2], and Manydepth[3]. There are three reasons for the selection of these models: (1) Efficient collection of training data is possible through the use of RGB cameras that capture monocular videos or stereo pairs, allowing for cost-effective model training; (2) The three models, among self-supervised MDE models, have demonstrated strong representativeness by achieving good performance using different frameworks; (3) The models are openly accessible in GitHub repositories and have an impact on the field or technical community.

**Training Settings.** In the projectable modeling study, a dataset comprising 1,200 paired samples is gathered under various environmental lighting conditions (e.g., shadow occlusion, twilight, indoor lighting, and low-light nighttime) and different backgrounds (e.g., gray, red, blue, and white), both with and without projection. This dataset is utilized to create a projectable list. Subsequently, we employ real-world driving scene data from KITTI dataset [12] to train and test adversarial patches. In Equation (6), the prior hyperparameter $\alpha_1$ and $\alpha_2$ are both set to 100. The dataset comprises calibrated stereo videos captured by LiDAR, encompassing various scenarios like highway driving, urban roads, rural roads, and congested traffic situations such as following vehicles and waiting at traffic lights. $\alpha$ and $\beta$ are set as 1.5 and 1, respectively, acting as hyperparameters to balance distinct loss components in Equation (16). All experiments are conducted on a server that is equipped with an NVIDIA GeForce RTX 3090-24G, Python 3.7, and PyTorch 1.5. We employ Adam optimizer with a learning rate of $10^{-3}$, a batch size of 8, and conduct training for 40 epochs.

**Physical Deployment.** We conduct physical attack experiments on campus roads and parking lots. Figure 6 presents the deployment of a physical attack in a parking lot, wherein the attacker utilizes a portable projector (Lenovo T6X) to project pre-trained patches onto specific target areas. Simultaneously, an iPhone 12 Pro functions as the victim camera, capturing dynamic scenes in the foreground.

**Evaluation Metrics.** To evaluate the attack performance of DepthCloak, the evaluation metrics include the attack success rate (ASR), root mean square error (RMSE), absolute relative error (AbsRel), root mean squared error in log-space (RMSE(log)), and

squared relative error (Sq Rel). The ASR is defined as the percentage of frames in which the average depth estimation error within the target area, in comparison to the ground truth (depth estimation without any attack), surpasses seven meters. This threshold of seven meters is considered crucial as it represents the minimum distance required for objects to be detected to prevent collisions in typical driving situations [2]. RMSE, AbsRel, RMSE(log), and Sq Rel represent the standardized metrics employed within KITTI dataset to assess the depth estimation errors.

### 5.2 Attack effectiveness

This section evaluates DepthCloak from three aspects: digital-world, physical-world, and the physical attack success rate.

**Digital-world.** Adopting the methodologies suggested by Yamanaka et al. [32] and Cheng et al. [1], adversarial patches are developed for three SOTA MDE models. The pre-trained patches are subsequently administered in a paste-like consistency onto the designated target regions. To guarantee effective concealment and practicality, the size of the patch is restricted to 20% of the total area of the target region. Subsequently, the depth estimation is performed by applying the target depth estimation models (utilized during training) to the digital world attack scenes, and the depth information generated by the models is gathered. The degree of misdirection induced by the patches is assessed by comparing the depth estimation results pre- and post-attack. DepthCloak simulates the projection of pre-trained patches onto designated areas within the digital world, with the resulting attack scenarios being provided as feedback to the target depth estimation models. 100 scenes from KITTI dataset are chosen for evaluation. Figure 7 compares the effects of patch pasting and digital simulation projection attack implementation methods.

Figure 7 depicts the results of estimating the depth using three models in clean scenes and following various adversarial attack strategies. The first row specifically displays the depth estimation results produced by these three MDE models in an uncontaminated, clean environment. The second and third rows display the output results of each depth estimation model following the implementation of adversarial attack strategies suggested by Yamanaka et al. [32] and Cheng et al. [1], correspondingly. The fourth row illustrates the results of depth estimation following the implementation of our DepthCloak. The presence of DepthCloak attacks in the digital domain can greatly distort the accuracy of depth estimation in the

---

[1]Monodepth2, https://github.com/nianticlabs/monodepth2
[2]Depth Hints, https://github.com/nianticlabs/depth-hints
[3]Manydepth, https://github.com/nianticlabs/manydepth

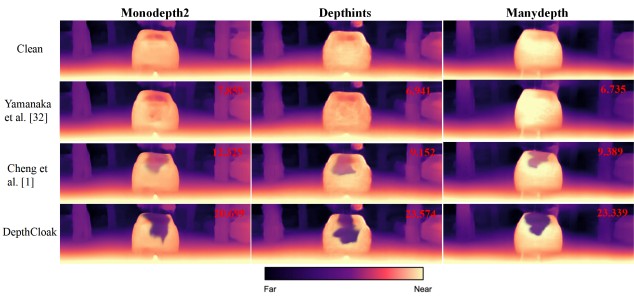

**Figure 8: Depth discrepancy under different attacks in physical-world.**

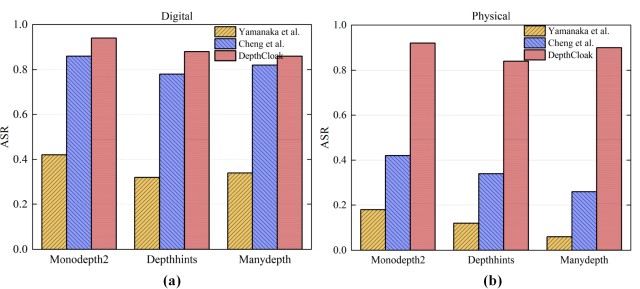

**Figure 9: Comparison of ASRs under different attacks.**

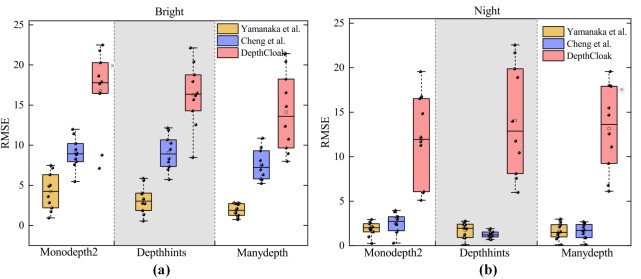

**Figure 10: Comparison of attack robustness in both bright and night lighting conditions.**

target area, leading to diverse levels of error among different models. In the case of the Monodepth2 model, DepthCloak has the potential to induce a depth estimation error of up to 26.409. Even when using the Manydepth model, which demonstrates relatively good performance, DepthCloak still causes a depth estimation RMSE of 20.742. Experiments conducted in 100 evaluation scenarios reveal that DepthCloak induces depth estimation RMSE of more than 15 across 76 scenes in the three designated depth estimation models. This phenomenon can be attributed to DepthCloak, which consists of highly customized projection patches designed for various scenarios, demonstrating effective interference capability.

**Physical-world.** In physical-world experimental, we utilize adversarial patches pre-trained using three attack strategies to evaluate their impact on three target depth estimation models. Throughout the experiment, the adversarial patches are projected onto a specified area located at the rear of a car. To document this procedure comprehensively, an 8-second video is captured utilizing an iPhone 12 Pro camera for each scenario, configured with a frame rate of 30 frames per second. Simultaneously, for comparative purposes, scenarios that are not affected by any attack are documented as the ground truth. Figure 8 illustrates the errors in depth estimation resulting from various attack strategies and target depth estimation models.

In Figure 8, DepthCloak demonstrates exceptional attack performance in real-world scenarios. When implementing the approach suggested by Cheng et al. [1] in the real world, there is a decrease in its attack effectiveness. While the approach suggested by Yamanaka et al. [32] demonstrates only slight variations in depth error during the shift from the digital domain to the real world, its overall

efficacy is comparatively limited. The primary factor contributing to this performance gap is rooted in the tailored and optimized design of DepthCloak for physical-world projection effects, in contrast to the other two methods which do not adequately account for the unique demands of transitioning attacks from the digital realm to the physical domain during both the design and training phases. Subsequent examination reveals that, when utilizing Depth-Cloak, approximately 70% of frames in the video recordings aimed at the three chosen target depth estimation models exhibit depth estimation RMSE exceeding 15.

**ASR.** In subsequent experiments, we assess the success rates of attacks using three strategies on three depth estimation models in digital and physical environments. In Figure 9, the results of experiments conducted in digital-world, which involve the analysis of 100 evaluation scenarios, indicate that the approach advocated by Cheng et al. [1] attains an ASR exceeding 78%. However, DepthCloak demonstrates superior performance in this particular environment. Transitioning to the physical domain, the strategies suggested by Yamanaka et al.[32] and Cheng et al. [1] demonstrate a decline in effectiveness. In contrast, DepthCloak demonstrates an ASR exceeding 80% even within physical environments. The series of experimental results present in Figure 9 not only emphasize the effectiveness of DepthCloak in navigating both digital and physical worlds but also demonstrate its sophisticated optimization considerations in design and implementation. These aspects enable the strategy to consistently achieve a high ASR across various environments.

## 5.3 Attack Robustness

To comprehensively assess the effectiveness of various adversarial patches in real-world application settings, we randomly select 10 cars with diverse background colors as subjects for experimentation. While maintaining a stationary camera position and a consistent distance to the target vehicle, the recording captures attack scenes in which various adversarial patches are projected onto specific areas of cars. This is conducted in both well-lit bright environments and low-light nighttime conditions. Furthermore, to establish a precise comparison baseline, scenes are also recorded without patch projection to serve as ground truth.

Figure 10 illustrates a comparison of the depth estimation errors caused by three attack strategies across different lighting conditions. Regardless of the lighting conditions, DepthCloak exhibits superior

**Table 1: Exploring ablation studying across diverse constraints.**

| Projectable Loss | Depth Model | Style Transfer | RMSE | AbsRel | RMSE(log) | Sq Rel |
|---|---|---|---|---|---|---|
| w/o | w/o | w/o | 5.356 | 0.132 | 0.229 | 0.877 |
| w | w/o | w/o | 8.024 | 0.118 | 0.275 | 1.574 |
| w/o | w | w/o | 9.452 | 0.105 | 0.293 | 1.615 |
| w/o | w/o | w | 5.981 | 0.128 | 0.220 | 0.953 |
| w | w | w/o | 18.325 | 0.226 | 0.361 | 3.524 |
| w | w/o | w | 9.859 | 0.157 | 0.242 | 1.867 |
| w/o | w | w | 12.537 | 0.262 | 0.314 | 2.916 |
| w | w | w | **20.699** | **0.334** | **0.455** | **5.311** |

attack performance in comparison to the other two methods. In well-illuminated settings, DepthCloak induces higher concentrations of errors in depth estimation. In nocturnal settings, DepthCloak exhibits a heightened advantage, whereas the efficacy of the other two offensive tactics is diminished. The results suggest that DepthCloak demonstrates significant adaptability and robustness to variations in environmental lighting conditions.

## 5.4 Ablation Study

**Determining Critical Parameter $\beta$.** We also assess the effect of different values of $\beta$ in Equation (16). The weight $\beta$ plays a crucial role in regulating the equilibrium between the content and style of adversarial patches. By maximizing the interference with the depth estimation model while simultaneously updating the visual appearance of patches, this weight aids in comprehending the target depth estimation model's sensitivity to variations in both style and content. Figure 11 shows the example of style patches across various parameters. It also showcases the depth estimation achieved by Monodepth2 following the application of these patches for attack. Additionally, it presents RMSE and structural similarity index (SSIM) comparing the patch with the reference style structure. The utilization of larger style transfer parameters has been noted to result in the creation of more inconspicuous adversarial patches; however, this may lead to a decrease in their attack effectiveness. This phenomenon occurs because larger style transfer parameters have a tendency to accentuate style characteristics, potentially leading to the loss or dilution of the original patch content. Additionally, the target model may exhibit reduced sensitivity towards these style attributes in comparison to content features. To strike a balance between attack performance and style, we set $\beta$ to 1 for all experiments.

**Component Analysis.** We conduct ablation experiments to understand the contributions of individual component modules in DepthCloak. This allows us to gain deeper insights into their impact on attack performance. These experiments systematically remove one or more modules from DepthCloak. For each modified configuration, Monodepth2 model is trained as the target model and validated using 20 scenes from KITTI dataset. Subsequently, we conduct a comprehensive evaluation of the effect of each configuration on attacks using five error assessment metrics.

The summarized results from the ablation experiments are presented in Table 1. Specifically, incorporating the projectable loss and depth modeling modules into the DepthCloak significantly

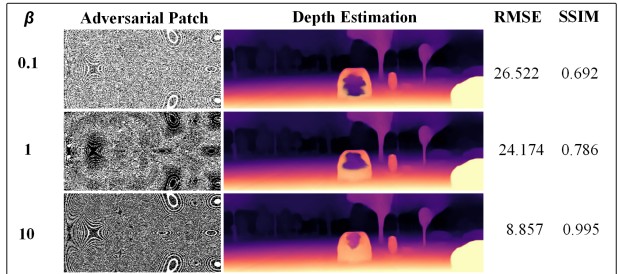

**Figure 11: The RMSE and SSIM of an attack example under different $\beta$.**

enhances attack performance. This discovery underscores the significant contributions of these two modules in collectively enhancing attack efficiency. Furthermore, when DepthCloak integrates all three modules, it attains the highest error across all five evaluation metrics. In contrast, the exclusive dependence on a singular module, such as projection loss, depth modeling, or style transfer, does not substantially improve attack performance. This suggests that the effect of an individual module in augmenting attack efficacy is constrained when lacking reinforcement from other modules. The coordination and synergy among modules play a crucial role in enhancing the effectiveness of attacks.

## 6 CONCLUSION

In this paper, we propose DepthCloak, an innovative optical projection attack tailored for monocular depth estimation models. To tackle the challenges arising from high-density color patch information, variations in target surface depth, and non-uniform pixel distribution in the physical world, we take several strategies. Firstly, we analyze the principle of "black-and-white light" projection in projectors and discover that this color distinction is formed by visual contrast differences. Subsequently, we transition the patch to a gray pattern to mitigate information loss during camera capture, attributed to the excessive complexity of patch colors. Finally, we introduce projection loss, depth projection modeling, and style transfer techniques into optical patch training. Extensive evaluation of large-scale datasets and physical vehicles validates the effectiveness and practicality of DepthCloak. In the future, we will explore a patch generation method based on diffusion models against multi-sensor fusion systems and provide corresponding defense strategies.

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
