# OpenReview forum: "DepthCloak: Projecting Optical Camouflage Patches for Erroneous Monocular Depth Estimation of Vehicles"
_acmmm.org/ACMMM/2024/Conference — MM2024 Poster_

### Official Review · Reviewer_4f4D · 2024-05-04

**Rating:** 4
**Confidence:** 1

**Summary:**

The authors introduce an adversarial optical patch to interfere with the MDE of vehicles. The experiments show that the proposed method can achieve depth errors of over 9 meters in both bright and night-time conditions while achieving an attack success rate of over 80% in the physical world.

**Strengths:**

The paper is well presented and clearly structured;

The experiments are thorough.

**Limitations:**

Since I am not an expert in the field of attack, I cannot accurately assess the novelty of this technique. I will therefore adjust my rating based on the opinions of other reviewers.

**Suitability:**

2

---

### Official Review · Reviewer_L4VS · 2024-05-22

**Rating:** 4
**Confidence:** 1

**Summary:**

In summary, this is a research paper that introduces a new optical projection attack method, aiming to deceive the vehicle-mounted monocular depth estimation system and provide new inspiration for related security and defense fields.

**Strengths:**

This work proposes a novel optical projection attack method called DepthCloak, which aims to interfere with monocular depth estimation (MDE) systems, an approach that has not been explored in existing research.

The method combines various techniques, such as GAN-based simulation, neighborhood averaging, and nonlinear mapping, to achieve more robust and effective adversarial patch generation.

Overall, this work demonstrates a high degree of innovation and technical sophistication, making an important contribution to the security defense field.

The achieved attack success rate exceeding 80% and depth estimation errors up to 9 meters demonstrate the practicality and effectiveness of the method.

Experiments conducted under both bright and night-time conditions showcase the robustness of the proposed approach.

**Limitations:**

The paper does not explore the applicability of DepthCloak in more complex scenarios, such as adverse weather conditions or complex road environments.

Future work could consider further optimizing the visual characteristics of the projection patches to enhance their naturalness and stealthiness.

Investigating detection and defense mechanisms against this type of optical projection attack could provide more solutions for security protection.

**Suitability:**

3

---

### Official Review · Reviewer_HYkS · 2024-05-25

**Rating:** 4
**Confidence:** 3

**Summary:**

The paper introduces an adversarial optical patch designed to interfere with the MDE of vehicles. which first simplifies the patch to a gray pattern because the projected “black-and-white light” has strong robustness to ambient light. Meanwhile, a generative adversarial network (GAN) based approach are proposed to simulate projections and deduce a projectable list. Then, the paper emploies neighborhood averaging to fill sparse depth values, compress all depth values into a reduced dynamic range via nonlinear mapping, and use these values to adjust the Gaussian blur radius as weight parameters, thereby simulating depth variation effects. Finally, by integrating Moiré pattern and applying style transfer techniques, the paper customizes adversarial patches featuring regularly arranged characteristics. Extensive experiments demonstrate that DepthCloak can achieve depth errors of over nine meters in both bright and night-time conditions while achieving an attack success rate of over 80% in the physical world.

**Strengths:**

1. For high visual information density, effect of non-planar surface and disparity in pixel distribution, the adversarial optical patch is novel.
2. The paper's methodology appears technically sound, with a logical progression from identifying the challenges to proposing a detailed solution framework.
3. Well-Organized and Clearly Written: The paper is structured in a logical manner, with clear sections dedicated to introducing the problem, detailing the proposed method, describing the experimental setup, and discussing the results.

**Limitations:**

1. The visualization comparison experiments are insufficient, with a limited number of comparison methods and depth prediction methods.
2. The paper lacks an analysis of the optimization process for the introduced hyperparameters. Could the authors elaborate on the process for optimizing these parameters?
3. In 2024, source code should be released and stated for scientific reproducibility.

**Suitability:**

2

---

### Meta-Review · Area_Chair_LzRW · 2024-07-01

**Recommendation:** Accept (Poster)
**Confidence:** 4

**Metareview:**

The proposed method has clear novelty as it introduces an adversarial optical patch designed to interfere with the MDE of vehicles. The performance of the proposed method has been sufficiently validated by extensive experiments conducted by authors. Furthermore, authors response to reviewers comment has verified the optimization process. Under the condition that authors have agreed to release the source code, there is a contribution to the community.